

# TaskHarbor
## System wspierający pracę zespołu programistów zgodny z metodyką SCRUM

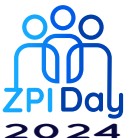

**Autorzy**: Marcin Moch · Mateusz Ptak · Dawid Zieliński · Mikołaj Czyżyk

**Opiekun:** Maciej Walczyński

**Streszczenie**

Celem projektu było opracowanie systemu wspierającego zarządzanie projektami zgodnie z metodyką SCRUM. System pozwala na efektywne zarządzanie zadaniami, bieżące monitorowanie postępów oraz retrospektywną analizę działań zespołu. Dzięki wykorzystaniu sztucznej inteligencji, narzędzie automatyzuje tworzenie opisów zadań, co zwiększa efektywność pracy oraz umożliwia bardziej precyzyjne planowanie. System charakteryzuje się intuicyjnym interfejsem, skalowalnością oraz wszechstronnością, umożliwiając zastosowanie zarówno w małych, jak i dużych zespołach. Projekt wdrożył nowoczesne technologie, takie jak Spring Boot, Next.js oraz AWS, aby zapewnić wydajność i elastyczność rozwiązania.

# 1 WSTĘP

## 1.1 Opis problemu

Problemem, który nasz projekt miał rozwiązać, była potrzeba usprawnienia pracy zespołów projektowych w szybko zmieniających się środowiskach. Istniejące narzędzia często ograniczają elastyczność lub wymagają skomplikowanej konfiguracji. Celem projektu było stworzenie prostego w obsłudze, a zarazem zaawansowanego systemu zgodnego z metodyką SCRUM, wspierającego efektywną organizację pracy zespołu.

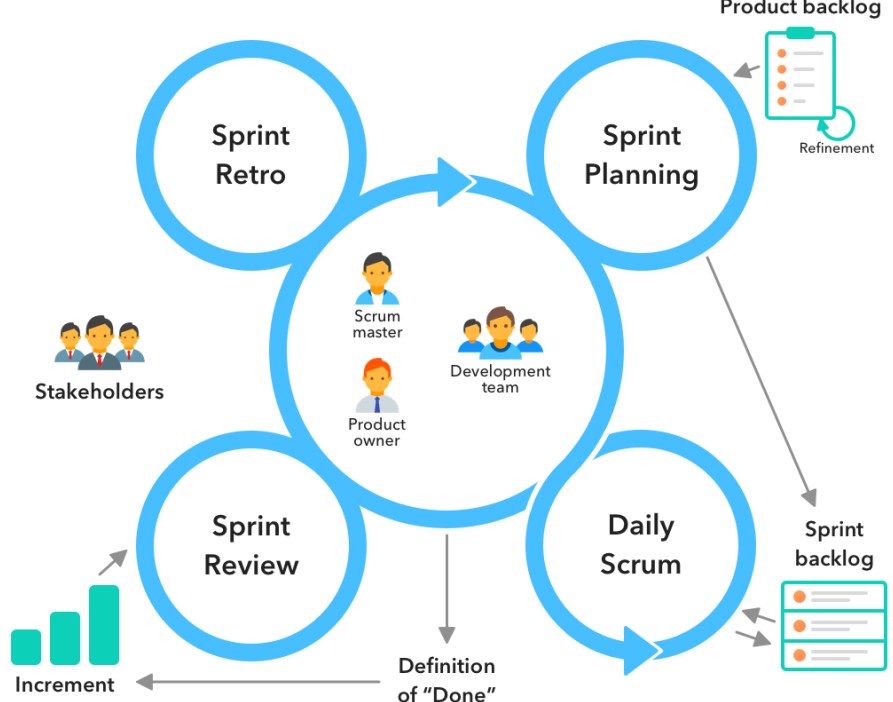

**Przebieg procesu SCRUM** [1]

## 1.2 Cele projektu

Głównym celem było stworzenie systemu, który:

· Ma przyjazny i intuicyjny interfejs użytkownika, skracający czas potrzebny na naukę obsługi systemu

· Ułatwia monitorowanie postępów oraz planowanie przyszłych zadań w podziale na sprinty i epiki

· Umożliwia współpracę wewnątrz zespołów w czasie rzeczywistym

· Automatyzuje opisywanie zadań i generowanie podsumowania retrospektywy za pomocą sztucznej inteligencji

· Jest skalowalny i obsługuje wzmożony ruch

· Kontroluje poziom uprawnień użytkowników za pomocą odpowiednich ról z podziałem na zespoły

· Wprowadza mechanizm retrospektywy, który umożliwia wyrażanie opinii na temat trwającego sprintu

## 1.3 Korzyści biznesowe i techniczne

Z perspektywy biznesowej projekt dostarcza istotną wartość, optymalizując pracę i eliminując część zadań w procesach biznesowych. Dzięki redukcji czasu poświęcanego na zadania manualne oraz automatyzacji powtarzalnych czynności, organizacje mogą znacząco obniżyć koszty operacyjne i zwiększyć efektywność swoich działań. Automatyzacja nie tylko podnosi produktywność, ale również uwalnia zasoby ludzkie, pozwalając zespołom skoncentrować się na strategicznych inicjatywach i kluczowych celach biznesowych. System usprawnia komunikację projektową wewnątrz zespołów, dzięki czemu przepływ informacji jest bardziej płynny i przejrzysty. Aktualizacja danych w czasie rzeczywistym wspiera podejmowanie świadomych decyzji, zwiększając nie tylko dokładność, ale również tempo realizacji projektów. Wyższy poziom przejrzystości procesów pomaga w skutecznym zarządzaniu zasobami, co przekłada się na lepsze wyniki biznesowe i większe zadowolenie klientów.

Z technicznego punktu widzenia, system został zaprojektowany z myślą o elastyczności i skalowalności, dzięki czemu może być wdrażany zarówno w małych startupach, jak i w dużych korporacjach. Bezpieczeństwo stanowi jeden z filarów rozwiązania – wykorzystanie jednokrotnego logowania (SSO). Chmurowa architektura oparta na zaawansowanych usługach AWS, takich jak Fargate i Aurora, zapewnia wysoką dostępność oraz niezawodność systemu. Konteneryzacja przy użyciu Dockera ułatwia wdrażanie nowych funkcji, minimalizując czas przestojów i pozwalając na szybkie skalowanie infrastruktury w odpowiedzi na rosnące potrzeby organizacji.

# 2 PODOBNE ROZWIĄZANIA

## 2.1 Przegląd istniejących rozwiązań

Współczesne narzędzia wspierające zarządzanie projektami zgodnie z metodyką SCRUM oferują szeroki wachlarz funkcji, które ułatwiają planowanie, organizację oraz monitorowanie pracy zespołów projektowych. Do najpopularniejszych rozwiązań należą Jira, Trello, Asana, Monday.com oraz ClickUp. Pomimo bogactwa funkcjonalności, większość z tych narzędzi opiera się na klasycznych tablicach Kanban lub wykresach Gantta, nie zawsze w pełni wspierając kluczowe elementy SCRUM, takie jak retrospektywy. Co więcej, narzędzia te rzadko wykorzystują sztuczną inteligencję w zaawansowany sposób do automatyzacji procesów, co w naszym przypadku stanowi obszar do dalszego rozwoju.

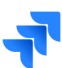 **Jira** [2] to jedno z najbardziej rozpoznawalnych narzędzi SCRUM, oferujące zaawansowane funkcje takie jak zarządzanie sprintami, tablice SCRUM oraz szczegółowe raportowanie postępów.

Jira zapewnia szeroką integrację z innymi systemami oraz możliwość konfiguracji dopasowanej do specyficznych wymagań zespołów. Niemniej jednak jej interfejs może być skomplikowany dla początkujących użytkowników, co wydłuża czas wdrożenia. Co istotne sztuczna inteligencja nie jest wykorzystywana do automatyzacji takich procesów jak analiza danych czy generowanie raportów.

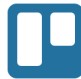 **Trello** [3] to narzędzie intuicyjne, bazujące na wizualnych tablicach Kanban, które umożliwiają łatwą organizację zadań i współpracę zespołową.

Trello jest szczególnie popularne wśród mniejszych zespołów ze względu na prostotę obsługi. Jednakże jego funkcjonalności ograniczają się głównie do zarządzania zadaniami, co czyni je mniej przydatnym w pełnym cyklu SCRUM, gdzie kluczowe są takie elementy jak retrospektywy, plany sprintów czy raporty końcowe.

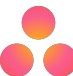 **Asana** [4] to narzędzie znane z elastyczności, które pozwala organizować zadania za pomocą tablic Kanban, harmonogramów i osi czasu.

Asana jest ceniona za możliwość dostosowania przepływów pracy do różnych potrzeb zespołów. Jednak brak wbudowanego wsparcia dla typowych rytuałów SCRUM, takich jak retrospektywy czy dedykowane raportowanie sprintów, czyni ją mniej efektywną w kontekście zarządzania zespołami stosującymi tę metodykę.

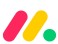 **Monday.com** [5] zapewnia szeroką funkcjonalność umożliwiającą zespołom dostosowanie przepływów pracy oraz wizualizację postępów w czasie rzeczywistym.

Monday.com wyróżnia się dużą elastycznością oraz szeroką możliwością integracji z aplikacjami firm trzecich. Jednak podobnie jak inne rozwiązania, narzędzie nie posiada wbudowanego wsparcia dla retrospektyw czy pełnego cyklu SCRUM, a funkcje automatyzacji są ograniczone do podstawowych reguł działania.

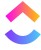 **ClickUp** [6] oferuje wszechstronne narzędzie, które łączy różne podejścia do zarządzania projektami, takie jak tablice Kanban, wykresy Gantta i listy zadań.

ClickUp wyróżnia się szerokimi możliwościami automatyzacji i personalizacji, co czyni je odpowiednim zarówno dla małych, jak i dużych zespołów. Jednak podobnie jak w przypadku innych narzędzi, wsparcie dla pełnego cyklu SCRUM, w tym retrospektyw czy dedykowanych funkcji dla sprintów, jest ograniczone i wymaga dodatkowej konfiguracji.

## 2.2 Wyróżniki projektu

Nasz projekt ma kilka kluczowych wyróżników, które sprawiają, że jest on bardziej efektywny niż tradycyjne narzędzia do zarządzania projektami:

- **Integracja modeli LLM do generowania opisów zadań:** Wykorzystanie dużych modeli językowych (LLM) pozwala na automatyczne generowanie opisów zadań oraz ich rozwijanie w oparciu o dane dostarczone przez użytkowników. System analizuje wprowadzone informacje, takie jak tytuły czy krótki opis, i na ich podstawie tworzy szczegółowe, spójne oraz zrozumiałe opisy, które od razu nadają się do wykorzystania w praktyce. Dzięki temu członkowie zespołu zyskują więcej czasu na realizację swoich obowiązków, a mniej na zajmowanie się szczegółową dokumentacją.

- **Uproszczony interfejs użytkownika:** Nasza aplikacja została zaprojektowana z myślą o użytkownikach o zróżnicowanym poziomie doświadczenia – od osób stawiających pierwsze kroki w pracy z narzędziami do zarządzania projektami po zaawansowanych specjalistów. Intuicyjny interfejs umożliwia łatwą i szybką nawigację, eliminując zbędne komplikacje i upraszczając codzienną pracę. Redukcja zbędnych opcji konfiguracyjnych oraz klarowne nazewnictwo funkcji pomagają skrócić krzywą uczenia umożliwiając szybkie wdrożenie systemu w organizacji.

- **Wbudowana funkcja retrospektywy:** Do każdego okresu pracy automatycznie generowana jest tablica retrospektywy, która umożliwia zespołom wymianę opinii na temat przebiegu projektów, atmosfery w grupie oraz efektywności współpracy. Tablica ta pozwala na zgłaszanie pozytywnych spostrzeżeń, wyzwań czy obszarów wymagających poprawy, co wspiera budowanie otwartej komunikacji i kultury feedbacku. Funkcjonalność ta wykorzystuje sztuczną inteligencję, która automatycznie przetwarza dane wpisane przez użytkowników. Funkcja pozwala na tworzenie list sugestii wynikających z wniosków retrospektywy, takich jak wdrożenie nowych praktyk czy eliminacja określonych problemów, co wspiera ciągłe doskonalenie procesów i budowanie silnych, dobrze zorganizowanych zespołów.

## 2.3 Założenia projektowe

Projekt oparty jest na kilku kluczowych założeniach technologicznych i projektowych:

- **Wybór technologii:** Do realizacji projektu użyto Spring Boot backend [7], Next.js (frontend) [8] oraz PostgreSQL [9] obsługujący bazę danych. Spring Boot pozwala na szybkie tworzenie rozwiązań backendowych. Next.js umożliwia tworzenie aplikacji frontendowych renderowanych po stronie serwera co zapewnia wydajność i ułatwia indeksowanie strony przez wyszukiwarki, co wpływa na wysokie pozycjonowanie serwisu. PostgreSQL jest natomiast stabilnym i rozbudowanym systemem zarządzania relacyjnymi bazami danych.

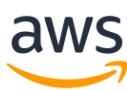 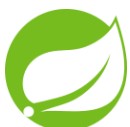 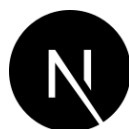 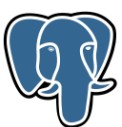 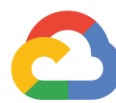 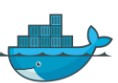

- **Skalowalność i elastyczność:** System jest zaprojektowany w sposób umożliwiający jego łatwą skalowalność i elastyczność. Konteneryzacja z użyciem Docker'a oraz rozbudowana, infrastruktura chmurowa w AWS [10], zapewnia wysoką dostępność i odporność na awarie, a także adaptację do zmieniającego się obciążenia aplikacji.

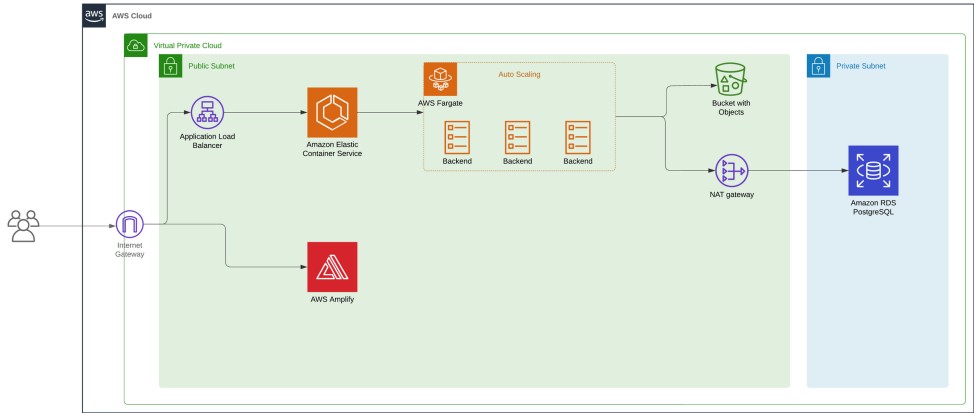

**Architektura aplikacji**

- **Ograniczenia czasowe:** Projekt był rozwijany w ramach wcześniejszego kursu podczas poprzedniego semestru, co pozwoliło na stworzenie podstawowego rozwiązania i wstępną implementację kluczowych funkcjonalności. Obecnie, w ramach bieżącego semestru, zespół kontynuuje prace, chcąc dodać dużą ilość nowych funkcjonalności oraz rozwijać projekt w sposób profesjonalny, z pełną architekturą opartą na AWS. Ze względu na ograniczenia czasowe wynikających z realizacji projektu w ramach semestru akademickiego, skupiono się na kluczowych aspektach, takich jak automatyzacja procesów, integracja z narzędziami zewnętrznymi oraz zapewnienie wysokiego poziomu bezpieczeństwa.

- **Ograniczenia finansowe:** Ze względu na fakt, że utrzymanie zawsze dostępnego środowiska testowego aplikacji w chmurze wiąże się ze sporymi kosztami, zaopatrzyliśmy aplikację, w dedykowaną część opartą całkowicie na serwisach AWS działających w trybie serverless (czyli dostępne zawsze, ale generujące koszty tylko za ich użycie). Pozwoliło to na zaimplementowanie cyklu życia aplikacji w chmurze, który uruchamia obrazy Docker'owe dopiero, gdy pojawia się ruch w aplikacji i zatrzymuje je, gdy aktywność ustanie, co pozwoliło w znaczący sposób ograniczyć koszta.

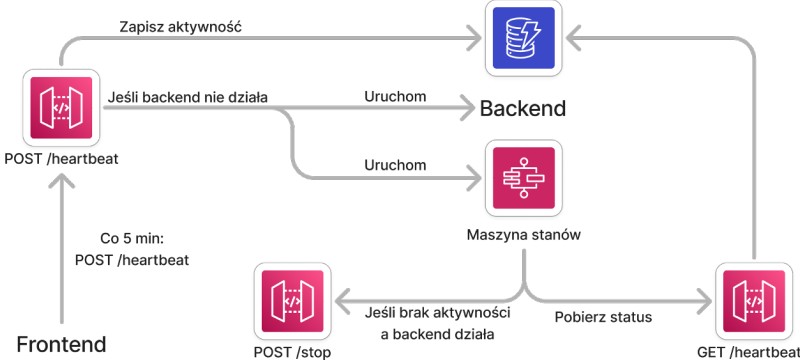

**Stan życia backendu**

# 3 WYNIKI

## 3.1 Funkcjonalności systemu

Nasz system wspiera cały proces zarządzania projektem zgodnie z metodyką SCRUM, oferując szereg kluczowych funkcji, które umożliwiają zespołom programistycznym i nie tylko, efektywne planowanie, monitorowanie postępów oraz retrospekcję. Do głównych funkcji systemu należą:

- **Rejestracja i logowanie użytkowników:** System umożliwia łatwą rejestrację nowych użytkowników oraz logowanie się do aplikacji z zachowaniem wysokich standardów bezpieczeństwa. System obsługuje różne metody autoryzacji, w tym logowanie za pomocą adresu e-mail i Google (SSO), co ułatwia szybki dostęp do aplikacji.

- **Powiadomienia e-mail:** Podczas zwykłej rejestracji użytkownicy otrzymują na adres e-mail link weryfikacyjny. System wysyła również powiadomienia e-mail o ważnych wydarzeniach, takich jak dodanie do zespołu, zmiana roli użytkownika, zapewniając, że członkowie zespołu są na bieżąco z postępami i zmianami w zespole.

- **Tworzenie i zarządzanie projektami:** Użytkownicy mogą łatwo tworzyć projekty, przypisywać je do zespołów oraz zarządzać strukturą pracy w ramach tych projektów.

- **Zarządzanie zespołami i ich członkami:** System umożliwia łatwe tworzenie zespołów, dodawanie członków oraz precyzyjne definiowanie ich ról i uprawnień. Role takie jak Product Owner, Scrum Master Developer czy zwykły User pozwalają na jasne określenie odpowiedzialności, co usprawnia organizację pracy i zabezpiecza dostęp do określonych funkcji systemu.

- **Zarządzanie sprintami, celami i przyrostami:** System umożliwia planowanie sprintów, definiowanie ich celów (Objectives) oraz grupowanie sprintów w przyrosty (Increments), co pozwala na lepszą organizację pracy i monitorowanie postępów w realizacji długoterminowych celów projektowych.

- **Tworzenie i zarządzanie zadaniami:** System umożliwia łatwe tworzenie zadań, przypisywanie ich do członków zespołu oraz organizowanie pracy w ramach sprintów. Zadania można również grupować w epiki, co pomaga w zarządzaniu większymi celami projektu.

- **Automatyczne generowanie opisów zadań oraz raportów retrospektywnych:** Dzięki integracji z technologiami sztucznej inteligencji (LLM), system automatycznie generuje opisy zadań, co przyspiesza proces ich planowania. Dodatkowo, na podstawie zgromadzonych danych, generowane są raporty retrospektywne, które podsumowują postępy zespołu, pomagając w analizie działań i wyciąganiu wniosków na przyszłość.

- **Wizualizacja postępów i analityka:** System oferuje różnorodne narzędzia do wizualizacji postępów, w tym diagramy przedstawiające zmiany stanów zadań w czasie, wykresy kołowe pokazujące liczbę zadań w poszczególnych stanach oraz *"Hall of Fame"*, który prezentuje osiągnięcia członków zespołu, takie jak liczba wykonanych story pointów. Dzięki tym funkcjom zespoły mogą na bieżąco monitorować swoje wyniki.

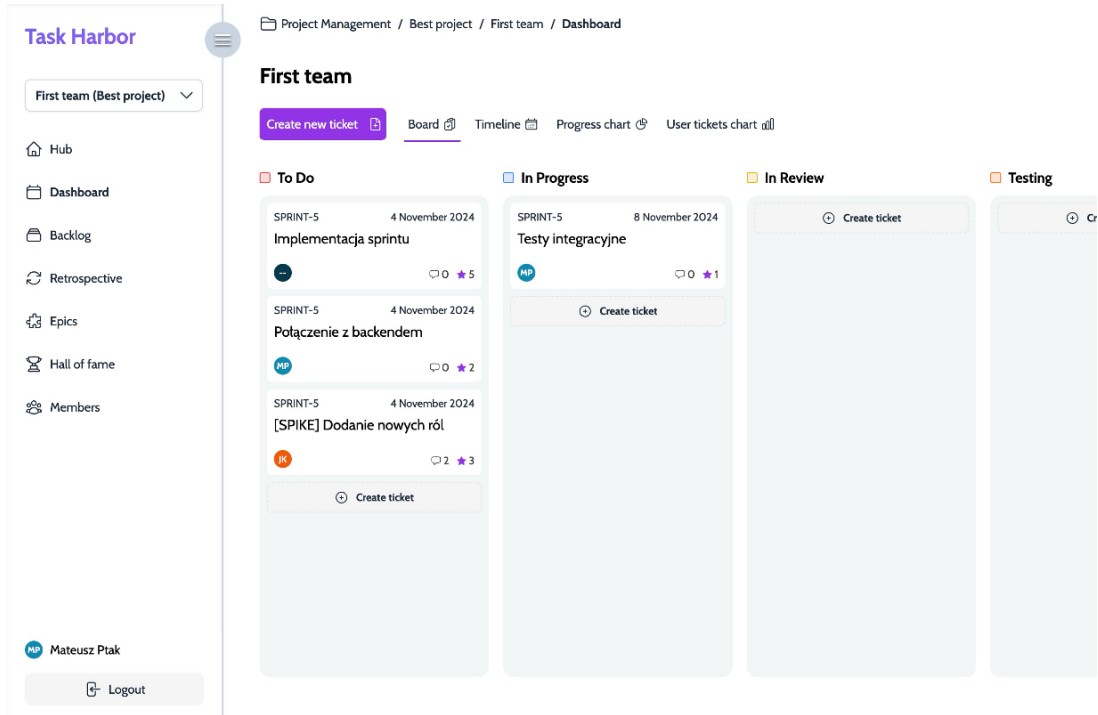

**Tablica zespołu**

## 3.2 Osiągnięte cele biznesowe i techniczne

- **Automatyzacja procesów i zwiększenie efektywności:** Dzięki automatyzacji wielu procesów, takich jak generowanie opisów zadań, raportów retrospektywnych zespół mógł skupić się na bardziej wartościowych zadaniach. Zmniejszenie czasu poświęcanego na czynności administracyjne pozwoliło na bardziej efektywne wykorzystanie zasobów.

- **Ulepszona komunikacja i przejrzystość interfejsu:** Wdrożenie systemu poprawiło komunikację w zespole dzięki prostemu, przyjaznemu i przejrzystemu interfejsowi użytkownika. Intuicyjny dostęp do zadań, postępów oraz nadchodzących terminów umożliwił szybkie przekazywanie informacji.

- **Optymalizacja zarządzania zadaniami i epikami:** Dzięki funkcjom umożliwiającym grupowanie zadań w epiki, zespół może lepiej zarządzać większymi celami projektowymi. Usprawnienie procesu zarządzania zadaniami w ramach sprintów pozwoliło na bardziej spójną realizację długoterminowych celów.

- **Lepsza analiza wyników i postępów pracy:** System dostarcza szczegółowych analiz w postaci wykresów i diagramów, takich jak zmiany stanów zadań w czasie czy "Hall of Fame", co pozwala na łatwiejsze śledzenie efektywności zespołu oraz szybsze identyfikowanie obszarów do poprawy.

- **Skuteczniejsze zarządzanie inkrementami:** System umożliwia organizowanie sprintów w ramach inkrementów, co pozwala na lepszą koordynację pracy zespołu i monitorowanie postępów w realizacji większych celów projektowych. Inkrementy stanowią logiczne przyrosty do realizacji kluczowych funkcji.

- **Zaawansowane zarządzanie rolami i uprawnieniami w zespole:** System pozwala na precyzyjne przypisywanie ról i uprawnień członkom zespołu, takich jak Product Owner, Scrum Master czy Developer. Dzięki tym funkcjom możliwe jest lepsze delegowanie odpowiedzialności oraz zabezpieczenie dostępu do kluczowych funkcji aplikacji, co zapewnia odpowiednią kontrolę.

## 3.3 Zastosowanie w praktyce

System jest w pełni skalowalny i elastyczny, dzięki czemu może być wdrożony w organizacjach o różnych rozmiarach. Jego modularna architektura pozwala na łatwą integrację z istniejącymi narzędziami oraz procesami. W szczególności:

- **Małe i średnie organizacje:** Dzięki prostemu interfejsowi oraz intuicyjnym funkcjom, system idealnie sprawdza się w małych i średnich zespołach, które potrzebują efektywnego narzędzia do zarządzania projektami bez konieczności skomplikowanej konfiguracji.

- **Duże organizacje:** Jego skalowalność pozwala na wdrożenie systemu w większych organizacjach, które korzystają z metodyk zwinnych na szerszą skalę.

- **Branże poza IT:** Chociaż system został zaprojektowany z myślą o zespołach programistycznych, jego uniwersalność sprawia, że może być zastosowany również w innych branżach, takich jak marketing, zarządzanie projektami budowlanymi czy inne sektory, które korzystają z metodyki SCRUM i potrzebują zwinnych narzędzi do efektywnego zarządzania projektami.

Dzięki łatwości wdrożenia, skalowalności oraz bogatej funkcjonalności, system może zostać szybko zaadoptowany w różnych środowiskach pracy, przynosząc korzyści zarówno w małych, jak i dużych organizacjach.

# 4 WNIOSKI

## 4.1 Podsumowanie wyników

Nasz projekt osiągnął założone cele, dostarczając zaawansowany system wspierający zarządzanie projektami w metodyce SCRUM. Dzięki zastosowaniu sztucznej inteligencji zautomatyzowano generowanie opisów zadań oraz raportów retrospektywnych, co znacząco przyspieszyło procesy planowania i analizy. Intuicyjny interfejs użytkownika, skalowalna architektura, pozwalają na łatwe wdrożenie rozwiązania zarówno w małych zespołach, jak i dużych organizacjach.

### 4.1.1 Najważniejszy sukces projektu

Największym sukcesem projektu było skuteczne wprowadzenie sztucznej inteligencji do codziennych procesów zarządzania projektami. Integracja modeli LLM umożliwiła automatyczne generowanie opisów zadań oraz raportów retrospektywnych, które są dostępne bezpośrednio w naszym systemie. Dzięki temu retrospektywa, jako jedno z kluczowych narzędzi do analizy postępów i wyciągania wniosków, stała się bardziej efektywna. Automatyzacja tych procesów przyczyniła się do zwiększenia efektywności pracy zespołów oraz poprawy jakości planowania. TaskHarbor, dzięki tym innowacjom, stanowi praktyczne narzędzie wspierające zwinne zarządzanie projektami, oferując natychmiastowy dostęp do raportów i retrospektyw, co ułatwia codzienne zarządzanie projektem.

## 4.2 Kierunki rozwoju

### 4.2.1 Możliwości rozwoju

- Integracja z narzędziami komunikacyjnymi, takimi jak Slack czy MS Teams, w celu usprawnienia współpracy zespołowej.

- Automatyzacja monitorowania repozytoriów kodu, integracja z GitHub oraz analiza commitów i pull requestów bezpośrednio w systemie.

- Implementacja funkcji generowania podsumowań i rekomendacji w oparciu o dane projektowe i historię sprintów, co umożliwi efektywniejsze planowanie przyszłych działań.

- Wdrożenie wsparcia dla wielojęzyczności, co pozwoli na adaptację systemu na rynkach międzynarodowych i poszerzenie jego dostępności dla użytkowników z różnych regionów świata.

## 4.3 Podziękowania

Serdeczne podziękowania kierujemy do naszego opiekuna naukowego, dr. Macieja Walczyńskiego, za cenne wskazówki i wsparcie merytoryczne. Chcielibyśmy również podziękować sobie nawzajem jako członkowie zespołu. Każdy z nas wniósł istotny wkład w projekt, wspierając innych w trudnych momentach, dzieląc się wiedzą i umiejętnościami oraz dbając o sprawiedliwy podział obowiązków. Dzięki wzajemnej pomocy i współpracy udało nam się zrealizować ten projekt z sukcesem.

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
