# OpenReview forum: "System wspierający pracę zespołu programistów zgodny z metodyką SCRUM"
_pwr.edu.pl/Wrocław_University_of_Science_and_Technology/2024/ZPI_Day — Wrocław University of Science and Technology 2024 ZPI Day Submission_

### Official Review · Reviewer_GHsc · 2024-12-03
**Raport o jakości typowej dla dobrych prac studenckich. Nie uwzględnia ilościowego potwierdzenia przyrostu jakości czy efektywności prowadzenia projektu poprzez zastosowanie LLM i zbudowanego systemu.**

**Confidence:** 4
**Significance Of Results:** 3
**Overall Quality:** 4

**Compliance With Template:**

4: High Quality – The article contains all the required sections, which are well-written and substantively correct, although minor errors or shortcomings may be present. The overall structure is clear and coherent.

**Description Of Results:**

4: High Quality – The results are described in detail and supported by usage examples or evaluations. The description is reliable but may lack full depth of analysis.

**Feedback On Consistency:**

>>> 1. Problemy językowe:

1.1 Język raportu jest poprawny a myśli wyrażone są w sposób zrozumiały. Pojawiają się jedynie nieliczne błędy, np. niepoprawne użycie znaków diakrytycznych w wypunktowaniu:

"Głównym celem było stworzenie systemu, który:
• Ma przyjazny i intuicyjny interfejs użytkownika, skracający czas potrzebny na naukę obsługi systemu
(..)
• Wprowadza mechanizm retrospektywy, który umożliwia wyrażanie opinii na temat trwającego sprintu"
=>
"Głównym celem było stworzenie systemu, który:
• Ma przyjazny i intuicyjny interfejs użytkownika, skracający czas potrzebny na naukę obsługi systemu,
(..)
• Wprowadza mechanizm retrospektywy, który umożliwia wyrażanie opinii na temat trwającego sprintu."

>>> 2. Problemy z prezentacją: Ilustracje nie są numerowane

>>> 3. Inne problemy:
3.1 Czy w sformułowaniu "Umożliwia współpracę wewnątrz zespołów w czasie rzeczywistym" pojęcie czasu rzeczywistego zostało użyte zgodnie z jego formalnym znaczeniem? Jeśli tak, to jakie przyjęto ograniczenia czasowe i jak analizowano ich spełnienie?
3.2 Wśród osiągniętych celów wymieniono "zwiększenie efektywności" oraz "poprawę jakości planowania". Brakuje określenia jak zmierzono zwiększenie efektywności i poprawę jakości planowania.

===EOT===

**Potential For Development:**

Yes

**Project Nature Evaluation:**

Yes, the project exhibits characteristics of an engineering work, with high level of utility, application of technical methods, and technological solutions.

**Technical Language Precision:**

3: Average Quality – The language is mostly appropriate but may contain minor terminological or stylistic errors. Some statements might lack precision or require improvement for better readability.

---

### Official Review · Reviewer_Ba7p · 2024-12-04
**Prezentowany projekt jest nowoczesnym systemem wspierającym zarządzanie projektami według metodyki SCRUM. Główne wyróżniki projektu obejmują integrację sztucznej inteligencji, intuicyjny interfejs oraz elastyczność wdrożenia w różnych skalach organizacyjnych. Opis projektu jest kompleksowy, logiczny i pokazuje potencjał narzędzia jako nowatorskiego rozwiązania dla zespołów projektowych.**

**Confidence:** 4
**Significance Of Results:** 4
**Overall Quality:** 5

**Compliance With Template:**

5: Very High Quality – The article contains all the required sections, which are written in a very detailed, clear, and error-free manner. The structure is professional and meets expectations, and the content adheres to the highest substantive and formal standards.

**Description Of Results:**

4: High Quality – The results are described in detail and supported by usage examples or evaluations. The description is reliable but may lack full depth of analysis.

**Feedback On Consistency:**

Opis projektu charakteryzuje się wysoką spójnością. Problem został jasno zdefiniowany, co podkreśla potrzeby zespołów działających w dynamicznym środowisku projektowym. Prezentacja wyników i funkcjonalności systemu jest logicznie powiązana z wyznaczonymi celami, a wnioski skutecznie podsumowują osiągnięcia. Każda część opisu, od analizy problemu po możliwości rozwoju, jest ze sobą powiązana, co sprawia, że treść jest przekonująca i czytelna.

**Potential For Development:**

Opis wskazuje na wiele obszarów dalszego rozwoju:
- Integracja z innymi narzędziami (Slack, MS Teams, GitHub).
- Funkcje analityczne i rekomendacyjne na podstawie historii projektów.
Te kierunki rozwoju wskazują na szerokie możliwości praktycznego zastosowania oraz komercjalizacji systemu w przyszłości.

**Project Nature Evaluation:**

Opis projektu jest szczegółowy, dobrze ustrukturyzowany i przedstawia praktyczne, inżynierskie podejście do problemu zarządzania projektami SCRUM. TaskHarbor to obiecujące narzędzie, które łączy nowoczesne technologie z realnymi potrzebami zespołów projektowych.

**Technical Language Precision:**

5: Very High Quality – The language is entirely appropriate for a technical report. All terms are used correctly and precisely, and the style is professional, clear, and coherent, without any errors or ambiguities.

---

### Official Review · Reviewer_uv6b · 2024-12-05
**Task Harbor, ZPI2024 review**

**Confidence:** 4
**Significance Of Results:** 4
**Overall Quality:** 3

**Compliance With Template:**

3: Average Quality – The article includes most of the required sections, but some may be incomplete, written in a general or unclear manner. The content is correct but requires further refinement.

**Description Of Results:**

3: Average Quality – The results are described with moderate detail. Some examples or evaluation elements are present but insufficiently developed or incomplete.

**Feedback On Consistency:**

W zgłoszonym artykule nie ma problemów ze spójnością. Autorzy zamieścili wszystkie wymagane sekcje, opisane są one w logicznej kolejności i opisują opracowaną aplikację. Niestety, język opisu jest w wielu miejscach (główne zarzuty poniżej) nieprecyzyjny, nie pozwalając na pełne zrozumienie nakładu pracy i osiągniętych wyników. W szczególności, w streszczeniu dowiadujemy się o wykorzystaniu sztucznej inteligencji, pierwsze doprecyzowanie tego jakie metody SI są wykorzystane występuje w Sect. 2.2, wskazując na integrację LLMów. Ani w wynikach, ani w podsumowaniu (wykorzystanie LLM jest wskazane w sekcji 4.1.1 jako największy sukces projektu) nie jest jednak ujawnione jakie modele zostały zintegrowane, czy finalnie jest ich więcej niż 1, czy użytkownik może lub musi skorzystać ze swoich credentiali podczas używania modułu LLM. Wobec braku tych informacji, trudno jest rzetelnie ocenić ten obszar prac za największy sukces. W ramach korekt po recenzjach warto sprecyzować podane informacje, co pozwoli w pełni odnieść się do nakładu pracy i osiągniętych sukcesów.
Innym przykładem braku precyzji jest wielokrotne powtarzanie frazy 'przyjazny i intuicyjny interfejs' lub zbliżonej, która nie jest mierzalna. Co więcej, autorzy nie zdecydowali się opisać procesu testowania interfejsu, czy to za pomocą walidatorów WCAG, raportu z subiektywnych testów użytkownika czy też kierowania się konkretnymi wytycznymi dotyczącymi ergonomii. Przeprowadzenie i opisanie takich testów spowodowałoby dokładniejsze zrozumienie wykonanych prac, możliwość weryfikacji osiągniętego rezultatu oraz (pod nieobecność linku czy to do systemu czy kodu źródłowego) pełne zobrazowanie sukcesów.
Ostatnia uwaga dotycząca formy prezentacji i zawartości rozdziałów dotyczy zamieszczonych rysunków. Pierwszym zarzutem odnośnie spójności, jest zamieszczanie rysunków w języku innym niż język artykułu. Jeśli z jakichś powodów, np. licencja, nie można wykorzystać zmodyfikowanego rysunku, takie objaśnienia powinny znaleźć się w podpisie (caption). Sam podpis powinien być rozbudowany, objaśniając czytelnikowi istotę rysunku oraz najważniejsze informacje z nim związane. Rysunki powinny być też numerowane, a dobrym zwyczajem jest wprowadzenie ich w tekście.
Uwzględniając limit stron, dodatkowe informacje (być może również link do kodów źródłowych) można zmieścić w dolnej części strony 2, rezygnując z wypunktowań oraz, jeśli to możliwe, przenosząc podziękowania do stopki.

**Potential For Development:**

W swoim artykule autorzy wskazali 4 kierunki potencjalnego rozwoju. Wskazane byłoby rozwinięcie tych pomysłów, określając (tam, gdzie to jest możliwe) sposób wykonania danych prac rozwojowych.
W przypadku komunikatora, być może (ze względu na licencję i przechowywanie informacji po stronie dostarczyciela) warto rozważyć integracje z narzędziami on-premise, np. Zulip.
Czy opisywany rozwój w kwestii analizy kodu i integracji z githubem to wykorzystanie pre-commitów i CI/CD zapewnianego przez GitHub, czy też rozwiązanie autorskie? W przypadku tego drugiego, warto opisać pomysł dokładniej, pozwalając czytelnikowi docenić włożony w fazę projektową nakład pracy.
Traktując dwa ostatnie punkty łącznie, być może warto wykorzystać już zintegrowane LLMy, wykorzystywane szeroko do podobnych działań, także w systemach konkurencyjnych (w zakresie generowania podsumowań).

**Project Nature Evaluation:**

Praca wykazuje się typowymi cechami inżynierskimi. Autorzy wskazują jasne zastosowanie oraz identyfikują lukę rynkową (w konkurencji pominięte są takie rozwiązania jak microsoft azure (board, ops), gitlab, bitrix, czy umożliwiające retrospektywę miro, pominięty jest też temat połączenia jiry z confluencem - związane jest to zapewne z limitem długości artykułu). Wobec braku precyzyjnego opisu opracowanego rozwiązania nie można rzetelnie oszacować szans wdrożenia efektów projektu, natomiast z pewnością ma on potencjał wdrożeniowy.
Uwzględniając uwagi zawarte w 'Feedback On Consistency' oraz zaprezentowany przez autorów opis konkurencji, sugerowałbym rozważenie przedstawienia integracji modułu do retrospektyw jako głównego osiągnięcia i skupienie się na dokładnym opisie jego możlowości.
Należy zauważyć, że w pracy nie zawarto żadnych mierzalnych wskaźników a nieprecyzyjne sformułowania nie pozwalają na rzetelną, inżynierską ocenę. W związku z tym należy stwierdzić, że opracowana aplikacja ma charakterystykę inżynierską, natomiast artykuł odbiega od prac inżynierskich i należy go doprecyzować.

**Technical Language Precision:**

3: Average Quality – The language is mostly appropriate but may contain minor terminological or stylistic errors. Some statements might lack precision or require improvement for better readability.

---

### Decision · Program_Chairs · 2024-12-10

Accept (Poster)